# The implementation and impact of non-invasive prenatal testing (NIPT) for Down's syndrome into antenatal screening programmes: A systematic review and meta-analysis

Elinor Sebire[1], Chithramali Hasanthika Rodrigo[1], Sohinee Bhattacharya[1], Mairead Black[1], Rachael Wood[2], Rute Vieira[1] *

1 Institute of Applied Health Sciences, School of Medicine, Medical Sciences and Nutrition, The University of Aberdeen, Aberdeen, Scotland, 2 Department of Covid-19 Health Protection Response, Public Health Scotland, Edinburgh, Scotland

* rute.vieira@abdn.ac.uk

**Data Availability Statement:** Most relevant data are within the manuscript and its Supporting

## Abstract

### Background

Non-invasive prenatal testing (NIPT) is a widely adopted maternal blood test that analyses foetal originating DNA to screen for foetal chromosomal conditions, including Down's syndrome (DS). The introduction of this test, which may have implications for important decisions made during pregnancy, requires continual monitoring and evaluation. This systematic review aims to assess the extent of NIPT introduction into national screening programmes for DS worldwide, its uptake, and impact on pregnancy outcomes.

### Methods and findings

The study protocol was published in PROSPERO (CRD42022306167). We systematically searched MEDLINE, CINAHL, Scopus, and Embase for population-based studies, government guidelines, and Public Health documents from 2010 onwards. Results summarised the national policies for NIPT implementation into screening programmes geographically, along with population uptake. Meta-analyses estimated the pooled proportions of women choosing invasive prenatal diagnosis (IPD) following a high chance biochemical screening result, before and after NIPT was introduced. Additionally, we meta-analysed outcomes (termination of pregnancy and live births) amongst high chance pregnancies identified by NIPT. Results demonstrated NIPT implementation in at least 27 countries. Uptake of second line NIPT varied, from 20.4% to 93.2% (n = 6). Following NIPT implementation, the proportion of women choosing IPD after high chance biochemical screening decreased from 75% (95% CI 53%, 88%, n = 5) to 43% (95%CI 31%, 56%, n = 5), an absolute risk reduction of 38%. A pooled estimate of 69% (95% CI 52%, 82%, n = 7) of high chance pregnancies after NIPT resulted in termination, whilst 8% (95% CI 3%, 21%, n = 7) had live births of babies with DS.

Information files. All data files used for analyses in this study are available in the project Open Science Framework database repository. URL: https://osf.io/85qg4/?view_only=179cbc7b4acc4a1cbbcf19d6b8dca3b5.

**Funding:** The author(s) received no specific funding for this work.

**Competing interests:** The authors have declared that no competing interests exist.

## Conclusions

NIPT has rapidly gained global acceptance, but population uptake is influenced by health-care structures, historical screening practices, and cultural factors. Our findings indicate a reduction in IPD tests following NIPT implementation, but limited pre-NIPT data hinder comprehensive impact assessment. Transparent, comparable data reporting is vital for monitoring NIPT's potential consequences.

## Introduction

Non-invasive prenatal testing (NIPT) was introduced as an antenatal screening test for Down's syndrome in 2011 [1], bringing with it the potential to impact existing screening pathways, reproductive choices, and medical care during pregnancy. Down's Syndrome (DS), or Trisomy 21, is the most common chromosomal condition seen in live births and results from a third (partial or complete) copy of chromosome 21 [2]. Screening for DS is part of routine antenatal care in many countries [3]. However, the offer of screening, its uptake by women, and women's diagnostic and reproductive choices following a high chance screening result, vary between countries and population subgroups, reflecting wider legal, religious, and cultural factors. It is often based on a combination of maternal age and blood markers such as alpha-feto protein, free beta-hCG and pregnancy associated plasma protein A [4]. Foetal nuchal translucency (NT) measurements from a routine first trimester ultrasound scan may also be used for women presenting for antenatal care early in pregnancy (S1 Fig). This combined approach, referred to as biochemical screening, gives a chance score for the likelihood of DS and has an average detection rate (DR) of 70% and a false positive rate (FPR) of 5% [4]. Subsequent diagnostic testing (genetic testing of foetal or placental cells obtained through amniocentesis or chorionic villus sampling, collectively referred to as invasive prenatal diagnosis (IPD), is used to confirm the screening results of high chance pregnancies. IPD is, however, associated with a 0.5% procedure-related risk of miscarriage [5].

Technological advancements in DNA sequencing methods have resulted in the introduction of NIPT as an antenatal screening test for DS. NIPT detects and characterises cell-free DNA (cf-DNA-NIPT), that is shed from the placenta into the maternal circulation, by using high throughput techniques such as chromosome-selective sequencing, massively parallel shotgun sequencing and single-nucleotide polymorphism based approaches [6–8]. These sequencing methods quantify the proportion of each chromosome in the maternal plasma and identify differences from the expected distribution. An excess or deficit in the expected chromosome profile would indicate a foetal trisomy or monosomy respectively [6, 9]. NIPT screening was first introduced commercially in Hong Kong and the US in 2011 and then marketed worldwide as a privately available test. It has since been recommended for implementation into national antenatal screening programmes by various governing bodies, including the American College of Obstetricians and Gynaecologists in 2012 [10] and the UK National Screening Committee in 2015 [1] as both a contingent, or 'second line', screen for high chance pregnancies identified by 'first line' biochemical screening [11–13], or as a first line test [14]. Previous reviews have reported on the widespread marketing and availability of NIPT testing [1], and on the considerable variability in population uptake of NIPT between countries—from 25–50% in the Netherlands, USA and Australia, to 75% in Belgium [15, 16]. However, these reviews do not focus solely on national-level implementation, and with NIPT use ever progressing, an updated understanding of the population-level impact of this test is required.

NIPT is more accurate than biochemical screening, with a DR of 99.2% and FPR of 0.09% [17]. The possibility of a false positive result means that NIPT is not diagnostic, and false positives can be indicative of placental mosaicism, vanishing twin, or maternal chromosomal abnormalities which could suggest a maternal cancer [18]. Therefore, as with biochemical screening, IPD is still required for a prenatal confirmation of positive NIPT results. However, by improving the accuracy of screening tests to identify women with a high chance of having a baby with DS, it is expected that the introduction of NIPT will reduce the number of unnecessary IPD tests performed [19]. Although a reduction has already been reported in some populations [15], it is important to understand if this trend is seen universally. Moreover, as the addition of this test changes the screening options and accuracy of information available to women during their pregnancy, there is potential to change the reproductive decisions made following the antenatal screening pathway. It could also inform pre and postnatal care, impacting the survival of babies with DS.

With new countries implementing NIPT into their national antenatal screening programmes every year [11, 16, 20, 21], continual monitoring is important for an up-to-date understanding of the extent of NIPT use in antenatal screening programmes for DS, along with its impact on IPD test uptake and pregnancy outcomes. Therefore, this systematic review will focus on two aspects of NIPT introduction into antenatal screening pathways for DS. Firstly, it will investigate the extent of NIPT use in government-implemented antenatal screening for Down's syndrome, including how NIPT has been implemented and its uptake by eligible women in these populations. Secondly, it will evaluate the impact of NIPT as a screening tool on specific reproductive choices and outcomes of pregnancies with DS in all clinical settings. This will include populations which may not have a national screening policy and will explore the proportion of those opting for IPD, termination of pregnancy and live births of babies with DS in eligible populations as key indicators of the impact of NIPT. This will provide necessary context for the future implementation of NIPT elsewhere and give insight into its potential impact on pregnancies of babies with DS.

## Methods

### Review questions

This systematic review aimed to address the following research questions:

A.  To what extent has non-invasive prenatal testing (NIPT) been implemented as part of a national antenatal screening programme for DS globally, and what is the uptake of NIPT in these populations?

B.  What impact has the use of NIPT had on the reproductive choices made and on pregnancy outcomes, in any clinical setting?

Terminology can be inconsistent; therefore, the glossary in Table 1 provides clarification and defines the terms to be used throughout this systematic review.

### Search strategies and selection criteria

Searches were conducted in MEDLINE, Embase, CINAHL and Scopus from January 2010 to March 2023, as the first reported use of NIPT for prenatal testing was in 2011 [1]. Two independent search inclusion and exclusion criteria were used to optimise screening and the identification of relevant studies for the two research questions presented.

**Part A (NIPT implementation in national antenatal screening programmes for DS globally).** Observational studies and healthcare or governmental publications were included,

**Table 1. Glossary of key terms.**

| Glossary | |
|---|---|
| *Antenatal screening programme* | For the purposes of this review this corresponds to screening tests offered to women during their pregnancy to screen for DS. |
| *High chance pregnancy* | Pregnancy that meets the locally decided threshold for having a higher chance of DS (chance > 1/X pregnancies). |
| *Biochemical testing* | Umbrella term for the combined use of maternal serum markers (first trimester: free beta human chorionic gonadotropin (HCG) and pregnancy associated plasma protein A (PAPP-A), second trimester: alpha-feto protein (AFP), free beta-HCG, inhibin-A and unconjugated oestriol), maternal age and nuchal translucency measurement on ultrasound to report the chance of the baby having DS. Also sometimes referred to as first trimester combined screening or traditional screening. |
| *First line screen* | A screening test that is usually offered to all pregnant women (may have inclusion/exclusion criteria) as their first test screening for the chance of the baby having DS. |
| *Second line screen* | A screening test offered to women who have already had one screening test e.g. biochemical, and have been identified as high chance. Can also be referred to as contingent screening. |
| *Pre-NIPT vs post-NIPT* | Time periods before and after NIPT was introduced into an antenatal screening programme for DS or offered in a clinical setting. For pre-NIPT, this may have been a period when biochemical screening was offered for DS, or no screening in some cases. |
| *Autonomous Region* | Any country/state/geographical region with autonomy to implement health policies for its population. |
| *Risk Threshold* | The limit set by a specific antenatal screening programme to define a high chance pregnancy with DS after biochemical testing (e.g., a chance of 1 in 150 pregnancies or higher). |

Key terms and definitions used throughout this review. These have been chosen to clarify definitions that are known to differ, and terms with variable names in this field.

from autonomous regions (countries/states/region) where NIPT had been implemented as part of national screening guidelines for DS. The specific terms used within this review are presented in the glossary. When there was evidence that a screening programme had been updated, or there were multiple papers covering the same population and period, the most recent and comprehensive study was chosen for inclusion. Studies were excluded when NIPT was not part of a national screening policy, it was not being used to screen for DS, no outcomes of interest were identified in the paper, or was a single centre study.

**Part B (impact of antenatal NIPT screening for DS on the prevalence of specific reproductive choices and pregnancy outcomes).** We included observational cohort studies and healthcare or governmental publications where NIPT was accessible as a screening tool, in any clinical setting, including single centre studies. NIPT did not need to have been implemented as part of a national policy. We included studies that provided data on the prevalence of IPD, terminations of pregnancy, and live births of babies with DS. Data on these outcomes before NIPT was introduced was also extracted where reported in that population.

For both parts, the search strategy (S1 Table) included a combination of keywords and MeSH headings relating to the terms 'non-invasive prenatal testing', 'Down's syndrome' and 'implementation'. Some studies were eligible for inclusion in both parts of this review. Reference lists and citations were searched in all included studies. Where reference to a national screening programme was found but the report was not in the included studies, handsearching of Google was used to try and identify the official report not returned from the database search. Authors of included studies were contacted if additional data were required for meta-analysis. Screening, data extraction, and quality assessment of at least 20% of the papers were carried out in duplicate by independent reviewers. No language restrictions were applied. Full details

of the search strategy can be found as part of the PROSPERO registered protocol (S1 File). This review was conducted in line with the PRISMA guidelines and checklist [22].

## Quality assessment

After full text screening, included studies were quality assessed using the Downs and Black quality checklist by one reviewer [23]. Papers were scored as follows: excellent (26–28), good (20–25), fair (15–19) and poor (<14), based on adaptations from evidenced-based healthcare centres by Hooper et al., [24].

## Data extraction and analysis

Study characteristics extracted for both parts, A and B, included the author and year of publication, study design and aims, time period of study, country or state, primary outcome measures of the paper, and declarations of interest/funding. Further specific outcomes extracted for each part of the review are described in Table 2. Extracted data was entered into an individual Excel spreadsheet file by each independent reviewer and then merged into a central file once any discrepancies had been agreed.

The data extracted for part A was summarised using appropriate graphical and descriptive analysis. A map of the regions that have implemented NIPT into a national screening programme was produced using Mapchart.net. Narrative synthesis was used to compare the implementation methods between regions. Uptake was compared between regions as a percentage of eligible women opting for NIPT where studies used comparable measures.

For part B, data was summarised using frequencies and percentages. Denominator data was the number of eligible women in the population for each stage of the pathway in each study (e.g. population eligible for screening). Data before NIPT was implemented (pre-NIPT) was used to provide comparison data where provided by the included studies.

Der Simonian and Laird random effects meta-analyses [25] were used to calculate pooled proportions with respective 95% CI and pooled odds ratios (OR), when there were at least three studies reporting an outcome in part B. Risk ratio and absolute risk difference was also calculated for clearer interpretation of the results. Where there was evidence of moderate heterogeneity ($I^2 > 40\%$), subgroup analyses, as defined *a priori*, and meta-regressions were used to assess heterogeneity-inducing factors, including risk threshold for NIPT, whether NIPT was

**Table 2. Outcomes of interest.**

| Part A | Part B |
|---|---|
| • Year of NIPT implementation.<br>• NIPT inclusion/exclusion criteria.<br>• When in the screening pathway NIPT is offered.<br>• How NIPT is funded (public/patient/both).<br>• Size of eligible population (n).<br>• Uptake of NIPT (% of eligible women who opt for NIPT). | • How NIPT is funded (public/patient//both)<br>• Average maternal age accessing NIPT.<br>• When NIPT is offered (first/second line screening)<br>• Number of high chance results after biochemical screening (when NIPT is offered as second line)<br>• High chance women after NIPT screening (first and/or second line)<br>• Number of IPD following high chance result from biochemical and/or NIPT screening.<br>• Number of terminations following high chance biochemical and/or NIPT screening.<br>• Number of live births of babies with Down's syndrome following high chance biochemical and/or NIPT screening.<br>• NIPT detection rate for DS and performance of NIPT in the screening pathway (sensitivity/specificity).<br>• Pre-NIPT implementation data on all variables above. |

Specific outcomes of interest extracted from the included studies for both parts A and/or B.

accessed as first or second line screening, uptake of NIPT and whether it was a national implementation (for studies included in both parts A and B). Publication bias was investigated using funnel plots. Not all papers reported both termination of pregnancy and live birth outcomes. Therefore, sensitivity analysis was performed on the subgroup of papers reporting both outcomes to investigate whether this influenced the estimated pooled proportions. Further sensitivity analyses, where each included study was omitted one by one, was performed for each of the outcomes meta-analysed, to identify the influence of individual studies on the pooled effect size and between-study heterogeneity. Meta-analyses results were displayed using forest plots. All analyses were conducted in R software (v4.1.3).

## Results

### Study selection

Database, citation and grey literature searching returned 1724 records after de-duplication, of which 167 studies underwent full text screening for inclusion in parts A, B, or both. The initial search looked for publications from 2010 to 10th May 2022, and was updated 29th March 2023. In total, 42 studies or reports were selected for inclusion in parts A and/or B. The stages of screening, and reasons for study exclusion at the full text screening stage are summarised in the PRISMA diagram (Fig 1). PRISMA checklist for systematic review reporting is provided in S2 File. The characteristics of the 42 included studies, including the quality assessment Downs and Black score, are described in detail in S1 Table.

### Part A–The implementation and uptake of NIPT in national antenatal screening programmes for Down's syndrome

Twenty-one manuscripts met the inclusion criteria for part A, reporting on 27 countries, states or autonomous regions in Europe, North America and Asia that implemented NIPT as part of a recommended antenatal screening policy for DS between 2011 and 2023 (Fig 2). Eight were retrospective cohort studies of population-based data [11, 13, 15, 20, 21, 26–28], evaluating the implementation of NIPT into their screening programmes for DS, and four were prospective cohort studies [14, 29–31]. Also included were eight official government documents describing the implementation of NIPT [12, 32–38] and one study that undertook a survey of clinical experts worldwide, providing data on multiple countries [16]. Survey estimates of NIPT uptake from this study were not extracted as it relied on best clinical estimates and was not based on population data.

Overall, the strategies for implementation of NIPT in antenatal DS screening have varied between populations–as both a first and second line screening test, publicly and privately funded, along with differing risk thresholds and criteria used to define high chance pregnancies for DS (Table 3).

### Risk of bias

The Downs and Black quality scores are presented in S1 Table. The studies that could be quality assessed were 'good' (n = 9) or 'fair' (n = 3). The government guidelines included in this review were not able to undergo risk of bias assessment as they did not follow the format of a scientific study.

**NIPT implementation.** The ways in which NIPT has been implemented as part of a prenatal screening policy for DS reported in our included studies are summarised in Table 3. NIPT has been implemented as a first (n = 7) or second (n = 15) line screening test. In Ontario (Canada) and Japan the option of either pathway is offered. Fig 3 depicts the first

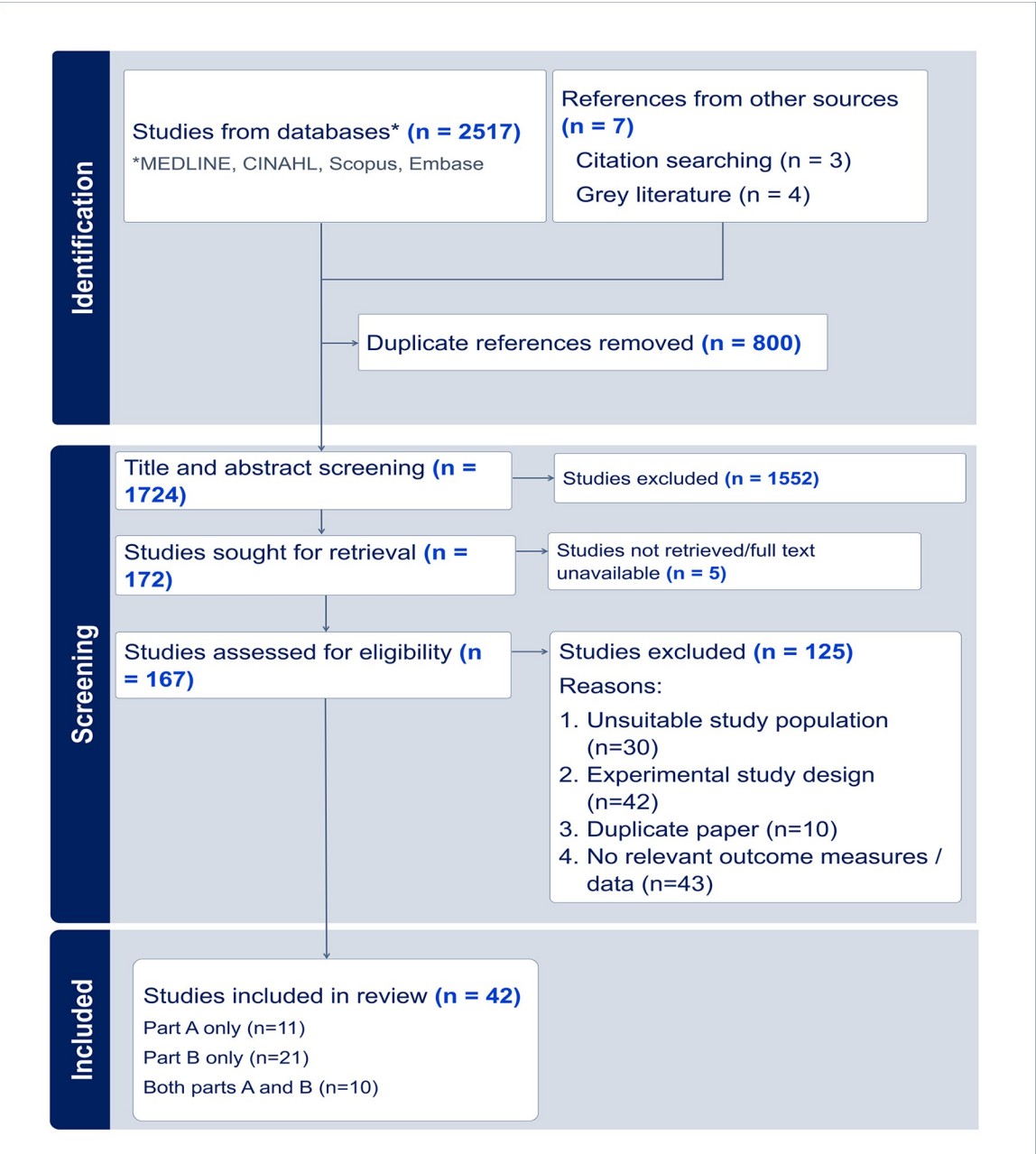

**Fig 1. PRISMA diagram.** Diagram to show the flow of screening stages and number of papers at each stage of the systematic review. It includes details from both the initial search up to 10th May 2022, and the updated search on 29th March 2023.

and second line screening pathways, as well as describing how the option of both is offered in some regions. Information on when NIPT is offered in the screening pathway was not available for Lithuania, Finland or Slovenia [16]. For second line NIPT screening, the risk threshold for NIPT access after a biochemical screening result varies considerably in the included studies; ranging from a chance over 1:1000 in France, Switzerland and Sweden, to a chance over 1:100 in Moscow. This threshold dictates the number of women eligible for

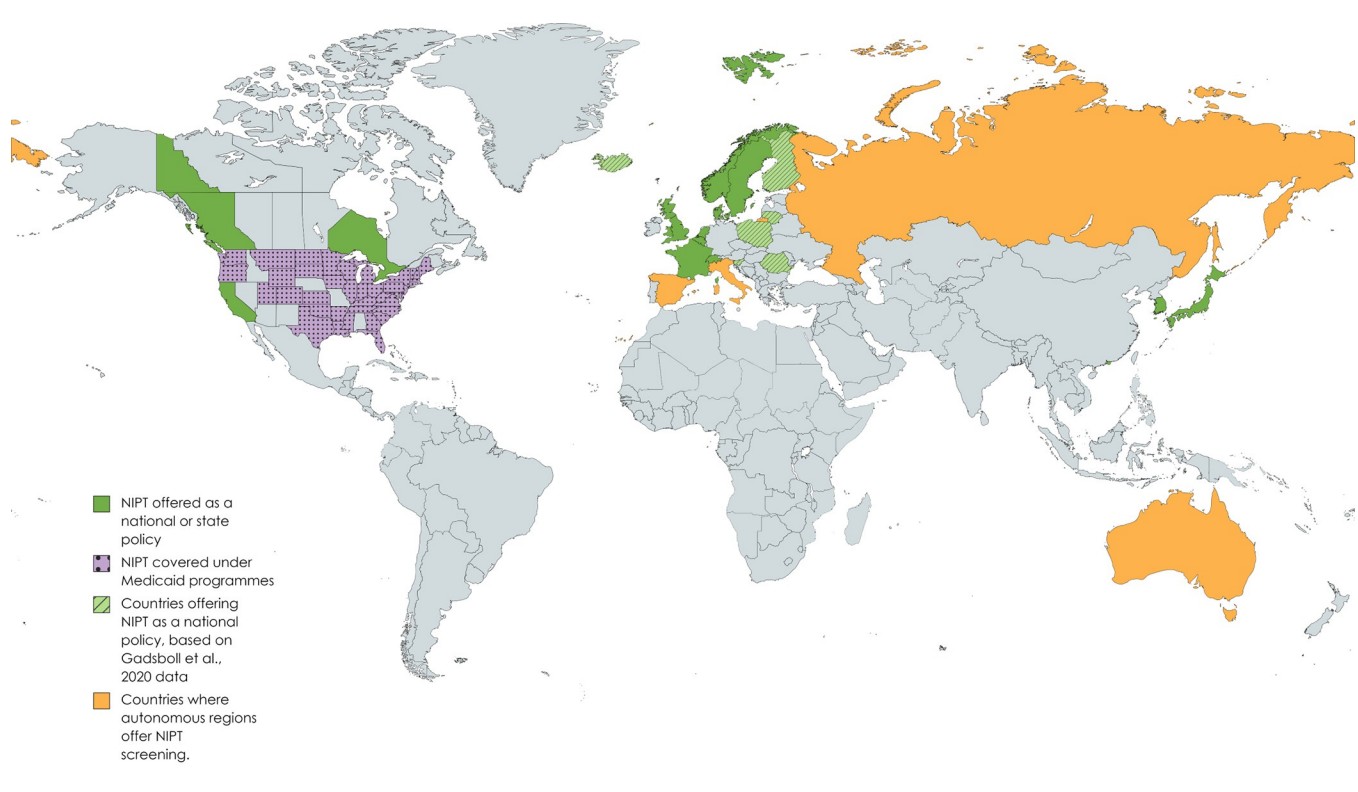

**Fig 2. Autonomous regions that have implemented NIPT into antenatal screening guidelines for Down's syndrome.** Presenting data from included studies in part A. This includes data from Gadsboll et al., [16] describing national NIPT implementation following survey responses by clinical experts (labelled in figure). Map created using Mapchart.net under a CC BY 4.0 license, with permission from Minas Giannekas, original copyright [2023]. NIPT = non-invasive prenatal testing.

NIPT and in turn determines the sensitivity, specificity, and overall cost-effectiveness of the screening programme.

Twelve of the 27 regions implementing NIPT reported that they provide NIPT at least partly funded by public health care or insurance plans, and four provide NIPT at full cost to the patient. The UK nations, except Northern Ireland, which has no DS antenatal screening programme, have implemented NIPT as a second line screening test for women with a higher chance of having a baby with DS [11, 12, 35].

Some countries updated their guidelines for implementation and access to NIPT in prenatal screening programmes throughout the period searched. The Netherlands pilot roll-out of NIPT in 2014 (named TRIDENT-1) offered funded NIPT as a second line screen for those pregnant women who had undergone first trimester combined testing (FCT) and were high chance for DS or had a positive medical history [39]. The guidelines were updated in 2017, with TRIDENT-2, an evaluative roll-out which offered NIPT to all pregnant women in the Netherlands as a first line screening test at a cost of 175 Euros [14]. This was then again updated with the end of the evaluation phase and the full launch of the programme in 2023. This demonstrates the dynamic nature of the changes occurring within NIPT screening policy for DS, even in the same population.

Poland, Romania, Lithuania, Finland, and Slovenia are reported by Gadsboll et al [16] to have implemented NIPT. However, our search did not return any documents relating to their screening programme.

**Table 3. The implementation methods of NIPT reported in the included studies of part A, when NIPT has been implemented as part of a national screening programme for Down's syndrome.**

| Country / state / province | Year of NIPT implementation | When is NIPT offered in the pathway? | Funding | Risk threshold for NIPT eligibility | Study ID |
|---|---|---|---|---|---|
| Hong Kong | 2011 | Second line screen | Patient funded | > = 1: 250 | Kou et al., 2016 [28] |
| Japan | 2013 | First and second line screen—depending on indications | Not reported | >1:300 classed as high risk–guidelines do not rely on this for access to NIPT | Samura et al., 2017 [40] |
| Victoria, Australia | 2013 | First line screen | Patient funded | N/A | Lindquist et al., 2019 [21] |
| Switzerland | 2015 | Second line screen | Publicly funded (insurance) | >1:1000 | Swiss public health insurance guidelines, 2015 [36] |
| Taiwan | 2015 | First line screen | Patient funded | N/A | Hsiao et al., 2022 [27] |
| Ontario, Canada | 2016 | First and second line screen | Publicly funded (insurance) | After first trimester screening > = 1:350 or second trimester screening > = 1:200 | Dougan et al., 2021 [26] |
| Sweden | 2016 | Second line screen | Not reported | 1:51–1:1000 | SFOG guidelines 2016 [38] |
| Belgium | 2017 | First line screen | Publicly funded (insurance) | N/A | Van den Bogaert et al., 2021 [15] |
| Denmark | 2017—Danish national guidelines for NIPT introduced in 2017, has been publicly available (funded) since 2013 in some regions of Denmark. | Second line screen | Public and patient funded–regional variations | Public setting–> = 1:300, some regions offer to intermediate risk: 1:300–1:700 or 1:1000. | Lund et al., 2020 [20] |
| South Spain–Andalucía | 2017 | Second line screen | Not reported | > = 1:280 | Torres Aguilar et al., 2021 [30] |
| The Netherlands | 2017—NIPT first introduced as TRIDENT-1 pilot scheme in 2014, updated in 2017 with TRIDENT-2 | First line screen (opt for either NIPT or FCT at similar price) | Subsidised (cost of 175 Euros unless previous history of DS) | N/A | Van der Meij, 2019 [14] |
| Yukon, Canada | 2017 | Second line screen (other indications e.g. twins / over 35 years offered NIPT as well) | Publicly funded (insurance) | Not reported | Health and social services Yukon, 2019 [33] |
| Wales, UK | 2018 | Second line screen | Publicly funded (NHS) | > = 1:150 | Bowden et al., 2022 [11] |
| Poissy Saint-Germain, France | 2019—Available under guidelines since 2015, fully funded from 2019 | Second line screen | Publicly funded (as of 2019) | 1:51–1:1000 | Duvillier et al., 2021 [13] |
| Poland, Romania, Iceland, Lithuania, Italy, Finland, Slovenia, USA | Not reported | Poland and Romania—second line screen | .Patient funded | 1:100–1:1000(invasive testing offered to those with a chance above 1:100) | Gadsboll et al 2020* [16] |
| | | USA—Second line screen. NIPT not offered under Medicaid for 9 States. | Medicaid coverage described for NIPT in many USA states | Not reported | |
| | | Iceland–Second line screen. Higher chance women can access NIPT over invasive testing if requested. | Publicly funded | Not reported | |
| | | Italy–Second line screen | Patient funded—Some regions publicly funded: Tuscany, Bolzano | Not reported | |
| Korea | 2020 | Second line screen | Not reported | > = 1:270 | Choe et al., 2021 [32] |

(*Continued*)

**Table 3.** (*Continued*)

| Country / state / province | Year of NIPT implementation | When is NIPT offered in the pathway? | Funding | Risk threshold for NIPT eligibility | Study ID |
|---|---|---|---|---|---|
| Moscow, Russia | 2020 | Second line screen | Not reported | > = 1:100, or 1:101–1:2500 | Olenev et al., 2021 [31] |
| Scotland | 2020 | Second line screen | Publicly funded (NHS) | > = 1:150 | Scottish Government—Chief Medical officer Directorate 2020 [35] |
| England | 2021 | Second line screen | Publicly funded (NHS) | > = 1:150 | Public Health England, 2021 [12] |
| California, USA | 2022 | First line screen | Publicly funded (insurance) | N/A | The California prenatal screening programme [37], Shah et al., 2014 [29] |
| Norway | 2022—Updated guideline offering universal NIPT for women over 35yrs. | First line screen | Publicly funded (insurance) | N/A | Norwegian Health Directorate—national professional guidelines [34] |

Implementation methods of NIPT for prenatal screening for Down's syndrome. Table describing the countries and autonomous regions that have implemented NIPT as part of a national prenatal screening programme for DS from part A included studies. The different aspects of implementation are summarised to understand the differences between implementation in different populations. Missing data is denoted by 'N/A'. References 26 and 34 both describe different aspects of the California prenatal screening programme.

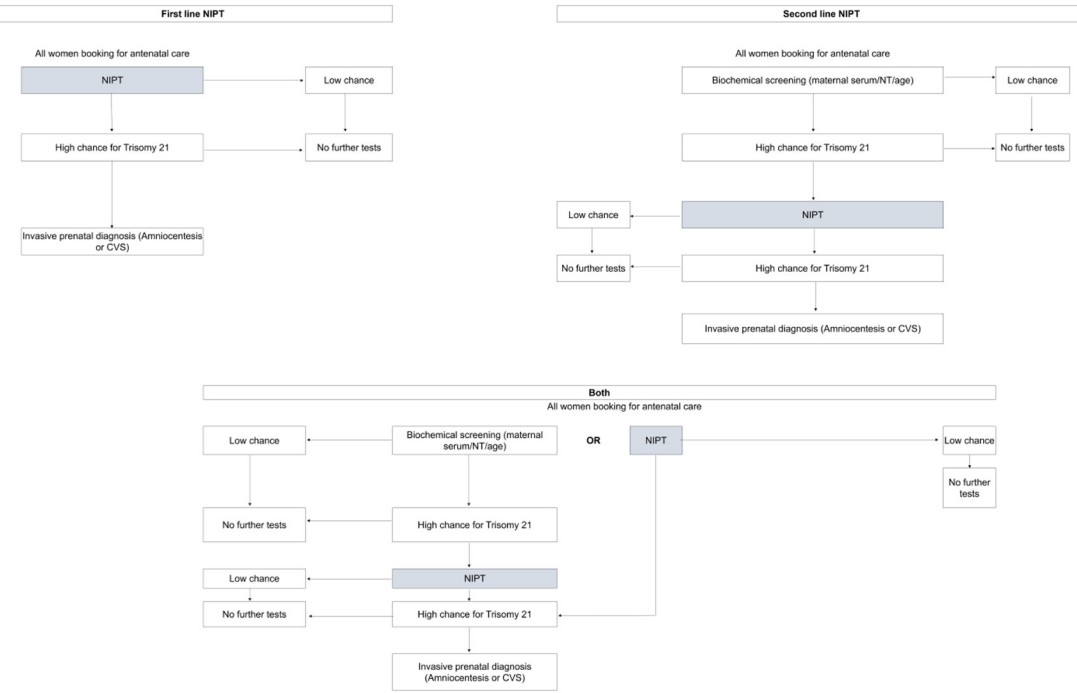

**Fig 3. Flow diagram for each pathway of NIPT implementation described in our included studies.** NIPT has been implemented as both a first or second line screen, in some cases the option of both is offered to those undergoing screening for Down's syndrome. NIPT = non-invasive prenatal testing, CVS = chorionic villus sampling, NT = nuchal translucency.

**Table 4. Uptake of NIPT for DS screening among eligible pregnancies in a population.**

| Country / state / province | Uptake (%) of NIPT among eligible pregnancies | | Reference |
|---|---|---|---|
| | **First line** | **Second line** | |
| **Andalucía, Spain** | | 93.2 (357/390) | [30] |
| **Belgium** | 78.7* (183621 / 115565) | | [15] |
| **California, USA** | | 30 (75/250) | [29] |
| **Denmark** | | 5.6* (3463/62198) | [20, 41] |
| **Hong Kong** | | 20.4 (107/401) | [28] |
| **Ontario, Canada** | 1.54 (5750/373682) | 73.9 (9901/13396) | [26] |
| **The Netherlands** | 42 (73239/173244) | | [14] |
| **Wales, UK** | | 84.3 (1073/1273) | [11] |

Grouped by first line and second line NIPT. For first line screening, this is the number of pregnancies at booking that opted for NIPT; for second line screening this is the number of higher chance women after biochemical screening that went on to have NIPT. Where indicated '*' uptake has been calculated as a percentage of all live births that had NIPT–for Belgium this % uptake will be close to the figure expected at antenatal booking because NIPT is offered first line, for Denmark the % uptake from antenatal booking is expected to be significantly different from the figure at antenatal booking, as second line, NIPT will only be available to those high chance pregnancies identified by biochemical screening. Average number of live births per year in Denmark from Sagi-Dain et al., 2021 [41]. First line screening may be lower in Ontario as women have the choice of both first and second line NIPT.

## Uptake of NIPT

The proportion of eligible women opting for NIPT was extracted from eight studies included in part A (Table 4).

The uptake of NIPT is highly variable between countries for both first and second line screening. First line NIPT is taken up by 1.54% of the women booking for antenatal care in Ontario, compared to 42% in The Netherlands, although this may be influenced by women having the choice of either first or second line screening in Ontario. Uptake of NIPT as a second line screen among women meeting the local risk threshold after biochemical screening ranges from 20.4% uptake in Hong Kong, to 93.2% in Andalucía, Spain.

## Part B–The impact of NIPT on IPD, termination of pregnancy and live births of babies with DS

Thirty-one articles were included for data extraction in part B, ten of which were also included in part A (S1 Table). Quality assessment classified the included studies into poor (n = 5), fair (n = 11) and good (n = 14). Data was extracted regarding pregnancy choices and outcomes (IPD, termination of pregnancy and live births with DS) from both single centres and populations where NIPT screening had been introduced. Where NIPT is offered as a second line screening test, the denominator of higher chance pregnancies following biochemical screening has been used where reported to enable comparison between pre and post-NIPT periods.

## Invasive prenatal diagnosis (IPD) following a high chance result for Down's syndrome after biochemical and NIPT screening

Twelve studies reported the proportion of women with a high chance pregnancy for DS that had IPD in the period after NIPT implementation, although only five of these reported outcome data for the pre-NIPT implementation as well.

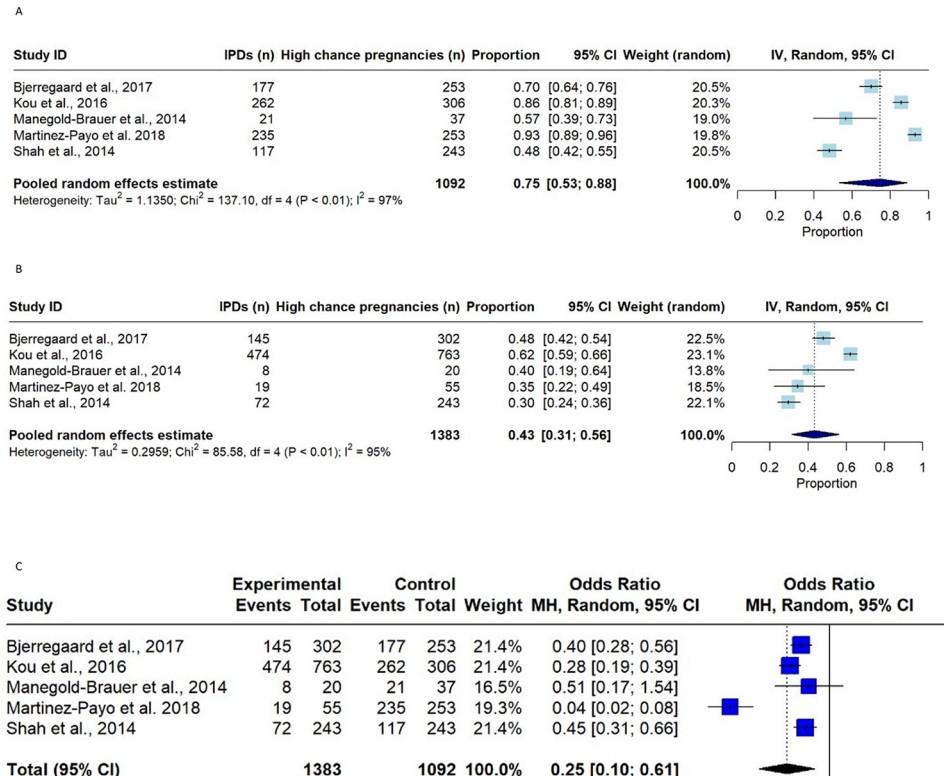

**Fig 4. Forest plots for meta-analyses of proportions of IPDs chosen by higher chance pregnancies after biochemical screening for DS.** A) pre-NIPT implementation B) post-NIPT implementation. C) Odds ratio meta-analysis, comparing the pre and post NIPT implementation time periods for the odds of having an IPD procedure in higher chance women.

**IPD uptake pre-NIPT implementation period vs post-NIPT implementation period.** Data was extracted from the five studies reporting the proportion of women opting for IPD pre-NIPT and post-NIPT implementation following a high chance biochemical screening result for DS.

In the pre-NIPT period, values ranged between 48% (117 out of 243) and 93% (235 out of 253) of pregnant women opting for IPD, who had a high chance biochemical screening result for DS. A meta-analysis of proportions was conducted to produce a random effects pooled estimate for the proportion of women with a high chance pregnancy opting for IPD of 75% (95% CI 53%, 88%, I2 = 97%) (Fig 4A) in the period before NIPT was introduced. After NIPT was available in these regions, 30% (72 out of 243) to 62% (474 out of 763) opted for IPD after a high chance biochemical screening result, with a pooled proportion of 43% (95%CI 31%, 56%; I2 = 95%) (Fig 4B). This suggests that the proportion of women opting for IPD following a high chance biochemical screening is reduced after NIPT is implemented as a second line screening test in the same population.

The pooled odds ratio (Fig 4C) suggests a significant reduction in the odds of opting for an IPD following a high chance biochemical screening result in the post-NIPT period compared to the odds of the pre-NIPT period (OR = 0.25;95% CI 0.1, 0.61; p = 0.0024; I$^2$ = 89%). This reduction can also be represented as a risk ratio of 0.62 (95% CI 0.55–0.70), and an absolute risk reduction of 38% (S3 File).

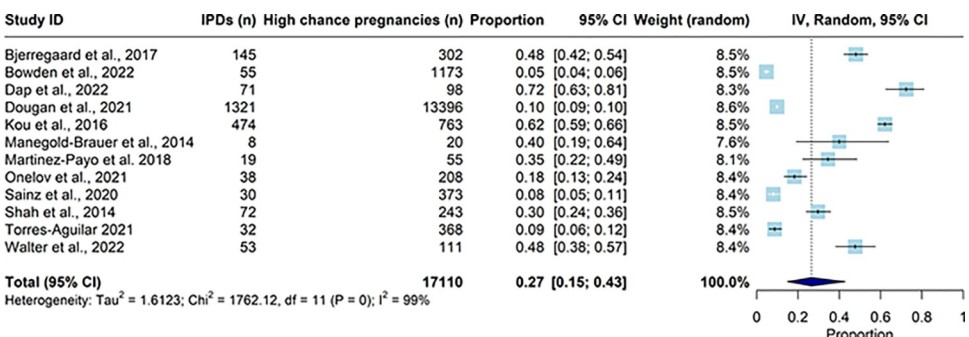

**Fig 5. Forest plot of the pooled estimate for proportion of higher chance pregnancies (resulting from biochemical screening) that went on to have an invasive prenatal diagnosis (IPD) after NIPT was implemented as a second line screen.**

**IPD uptake among pregnant women with a high chance biochemical screening result after NIPT implementation as a second line screening test.** To obtain a more robust estimate of the pooled proportion of women with a high chance pregnancy for DS, after biochemical screening, that chose IPD in the post-NIPT period, a meta-analysis was conducted using data from all studies (n = 12, Fig 5). The resulting pooled proportion showed that 27% (95% CI 15%, 43%; $I^2$ = 99%) of high chance pregnancies opted for IPD, when NIPT is available as a second line screening test.

**Uptake of IPD among pregnant women with a high chance NIPT result for DS.** The combined pooled proportion of women opting for IPD following a high chance NIPT result, when offered either as a first or second line screen (n = 21), is 87% (95% CI 80%, 92%; $I^2$ = 94%) (S3 File). The pooled proportion of IPDs opted for after a high chance NIPT result was also calculated separately for first and second line implemented NIPT (Fig 6A and 6B). The pooled proportion of those opting for IPD following a high chance NIPT result is higher when NIPT is introduced as a first line screening test versus second line introduction—89% (95% CI 80%, 94%; $I^2$ = 94%) of pregnancies, compared to 80% (95% CI 69%, 88%; $I^2$ = 90%)—however there is substantial overlap of the respective CIs. Meta-regression to adjust for first or second line offering of NIPT did not demonstrate any significant difference in IPD uptake between the groups (n = 21, p = 0.36).

**Subgroup and meta-regression analyses–factors influencing IPD uptake.** Subgroup and meta-regression analyses were used to explore the influence of specific variables on the heterogeneity present in the above meta-analyses investigating IPD uptake (Table 5). No significant difference was found between those studies where the biochemical screening threshold to determine high chance pregnancies for DS after biochemical screening (and subsequent access to NIPT) was more than 1:149 when compared to less than 1:150 (p = 0.22). Although, when repeated for high chance pregnancies identified by NIPT testing, there did seem to be a significant difference in IPD uptake between the population of pregnancies that accessed NIPT through either the two biochemical risk threshold groups and when no threshold was in place (first line NIPT) (p<0.0001). Whether NIPT is offered first, second line or as either, does not seem to have a significant effect on the proportion of women choosing to have an IPD after a high chance result from NIPT; subgroup analysis showed 90% (n = 8), 81% (n = 11) and 91% (n = 2) of high chance pregnancies went on to choose IPD when NIPT was implemented as a first, second or as both first- and second-line screening respectively (p = 0.28). Moreover, there was no evidence for whether NIPT was implemented as part of a national guideline or not influencing the proportion of women going on to have IPD (Table 5).

A

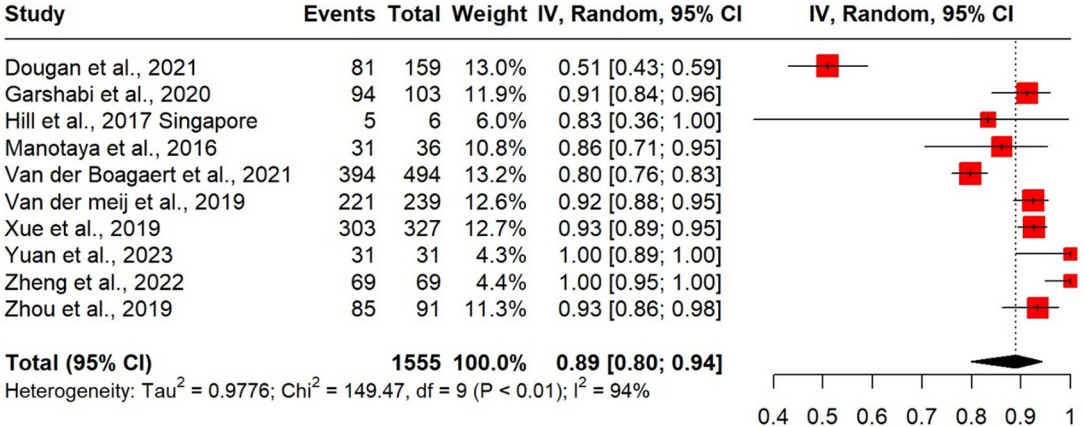

B

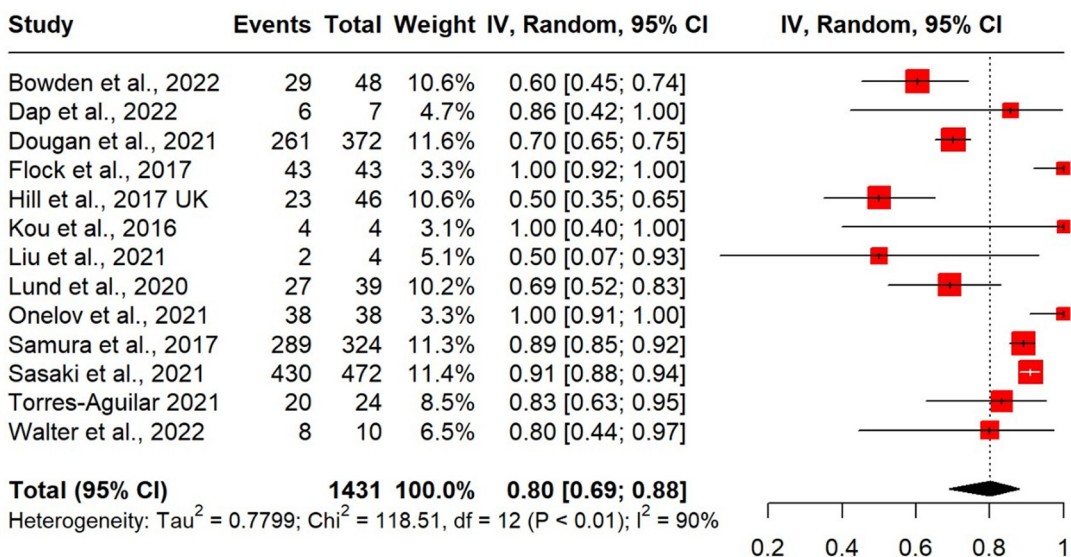

**Fig 6.** A) Forest plot of pooled proportion of women with a high chance NIPT result that opt for an IPD procedure (NIPT available as a first line screening test); B) Forest plot of pooled proportion women with high chance NIPT result that opt for an IPD (NIPT available as a second line screening test).

Results of subgroup and meta-regression analyses examining the associations between specific factors that might influence the proportion of high chance pregnancies (after biochemical or NIPT screening) for DS undergoing IPD, post-NIPT implementation. P value below 0.05 is considered significant.

Furthermore, meta-regression analysis adjusting for the uptake of NIPT when offered as a second line screen, in pregnancies with a high chance biochemical screening, showed a significant influence of NIPT uptake on IPD use (p = 0.0002). The regression coefficient (-0.044(95% CI -0.067, -0.021)) suggests that for every 1% increase in uptake of NIPT, the uptake of IPD

**Table 5. Subgroup and meta-regression analyses results.**

| Subgroup analyses | | |
|---|---|---|
| **Predictor variables** | **Proportion opting for IPD (95% CI) random effects** | **Test for subgroup differences (p-value)** |
| Risk threshold for access to NIPT testing (% opting for IPD out of women with a high chance *biochemical screening* result) | | |
| • Chance higher than 1:149 (n = 3) | 0.45(0.17,0.77) | 0.22 |
| • Chance equal to or lower than 1:150 (n = 5) | 0.19(0.06,0.48) | |
| Risk threshold for access to NIPT testing (% opting for IPD out of women with a positive *NIPT* result) | | |
| • Biochemical chance higher than 1:149 (n = 2) | 0.94(0.60,0.99) | <0.0001 |
| • Biochemical chance equal to or lower than 1:150 (n = 6) | 0.65(0.61,0.69) | |
| • No threshold for access (first line NIPT) (n = 9) i.e. *did the risk threshold for NIPT access after biochemical screening (or no threshold for first line NIPT) change the proportion of women opting for IPD?* | 0.90(0.86,0.93) | |
| • When NIPT is offered in the pathway (% opting for IPD out of women with a high chance NIPT result): | | |
| • First (n = 8) | 0.90(0.85,0.93) | 0.28 |
| • Second (n = 11) | 0.82(0.68,0.90) | |
| • First or Second (n = 2) | 0.91(0.25,0.99) | |
| Meta-regression analyses | | |
| **Predictor variable** | **Regression coefficient** | **P-value** |
| Uptake of NIPT as a second line test (%) (n = 9) | -0.0441(-0.067, -0.021) | 0.0002 |
| Uptake of NIPT as a first line test (%) (n = 4) | 0.0085(-0.024,0.04) | 0.60 |
| NIPT implemented as a national guideline (yes) | -0.377(-1.3,0.55) | 0.42 |

decreases by -0.044. The uptake of first line NIPT screening was not shown to be a significant factor influencing the proportion of IPDs (p = 0.60) (Table 5).

In summary, these analyses demonstrate that the uptake of IPD by high chance pregnant women can be explained in part by how NIPT has been implemented. Specifically, the risk threshold set, when NIPT is offered in a pathway, and the level of NIPT uptake as a second line test. However, this did depend on the high chance sub-group population taken (from either biochemical or NIPT screening) and none of these analyses were able to explain a considerable proportion of the heterogeneity, pointing to other potentially significant factors that we have been unable to account for using the available data.

**Termination of pregnancies following a high chance screening result for Down's syndrome.** Only one study provided data reporting the proportion of terminations of pregnancy for babies with DS pre-NIPT implementation. Bjerregaard et al. [42] reports 10 (3.95%) terminations of pregnancy (TOP) for DS after 253 high chance biochemical screening results.

Two studies provided data post-NIPT implementation on TOP for DS, after a high chance biochemical screening result, reporting proportions of 4.96% (15/302) and 10.2% (10/98) [42, 43]. Seven included studies provided data in the number of TOPs following a high chance first or second line NIPT result for DS. Meta-analysis of this data (n = 7) gave a pooled proportion of 69% (95% CI 52%, 82%; $I^2$ = 88%) opting for TOP (Fig 7).

**Live births following a high chance NIPT result for Down's syndrome.** None of the included studies reported the proportion of live births of babies with DS following a high chance biochemical screening test pre-NIPT implementation.

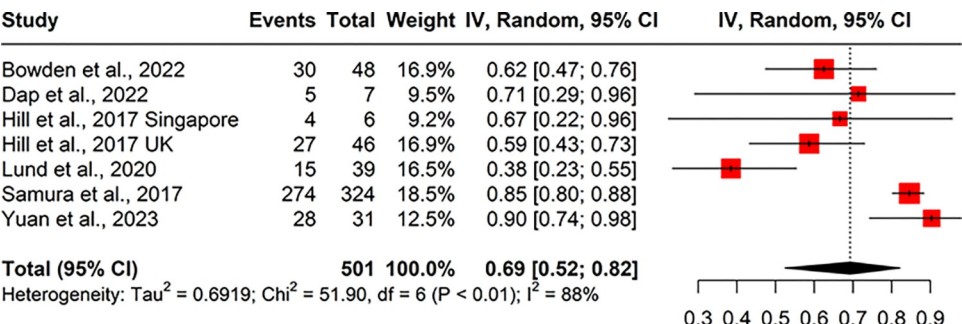

**Fig 7. Forest plot of the proportion of terminations of pregnancy following a high chance NIPT result for DS, in the post-NIPT period.**

In the post-NIPT period, Dap et al., 2022 reported two live births of babies with DS following a high chance biochemical screening result, out of 98 higher chance women identified by biochemical screening.

After a high chance NIPT result, seven studies reported the live births of babies with DS (Fig 8). The proportion of live births of babies with DS after a high chance NIPT result ranged between 0% (0/7) and 28% (11/39), with a pooled proportion of 8% (95% CI 3%, 21%; $I^2$ = 87%).

Only four of the studies included in the TOP and live birth analysis reported data on both outcomes, allowing a comparison of these estimates in the same population. The pooled proportions of terminations of pregnancy and live births were 72% (95% CI 50%, 87%; $I^2$ = 91%) and 10% (95%CI 3% to 28%; $I^2$ = 90%) respectively. There is some discrepancy between the values for live births and terminations of pregnancies, where we do not have an outcome for each pregnancy reported in some included papers. The remainder of pregnancies will have ended in spontaneous pregnancy loss or will have been lost to follow up within the study period.

**Sensitivity analyses.** Funnel plots of each meta-analysis demonstrated some asymmetry, which could be due to publication bias, or true heterogeneity within the data (S4 Appendix). Sensitivity analyses were run on each meta-analysis to explore heterogeneity, whereby one study was omitted at a time. None of the studies in the meta-analyses exploring IPD uptake showed significant influence over the estimated proportion or heterogeneity of the analysis. This was also found in the TOP meta-analyses, however, for the live birth meta-analysis omitting Samura et al. was seen to reduce the heterogeneity to 64% (from 87%) and gave a pooled

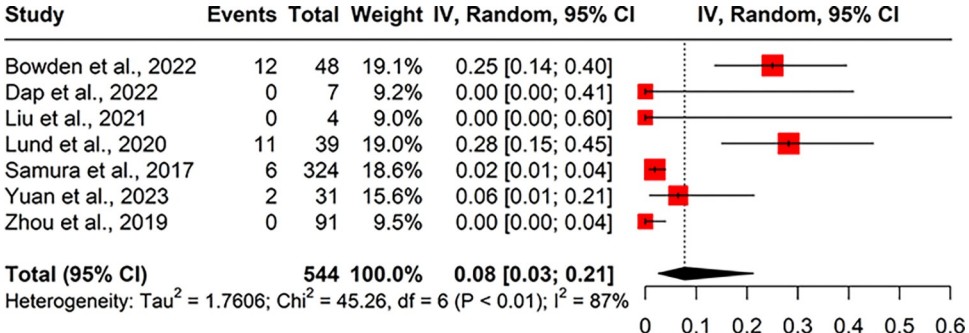

**Fig 8. Forest plot of meta-analysis of proportion for live births of babies with Down's syndrome born after a high chance NIPT result.**

estimate of 11% (95% CI 4%, 28%). S3 Appendix provides the results of each sensitivity analysis.

## Discussion

### Summary of findings

Overall, part A demonstrates huge variation in NIPT implementation and its uptake between eligible populations. NIPT has been implemented throughout the period of the search (2011–2023) as a first and second line screen in 27 autonomous regions, both funded by governments and privately. The differences in NIPT implementation could partially explain the heterogeneity in uptake. However, differences between healthcare systems, access to medical and social care resources, and societal attitudes towards termination of pregnancy (TOP) and other aspects of prenatal screening will also play a role in the acceptability of NIPT to the general public.

The overall proportion of IPD procedures after biochemical screening seems to have reduced after the introduction of NIPT as a second line screen. Moreover, pooled results showed that, overall, 89% of pregnant women opted for IPD after a high chance NIPT result, 69% chose to have a TOP, and 8% of high chance pregnancies after NIPT ended in live births of babies with DS, although with a high level of heterogeneity. When it was possible to do so, adjusting for different aspects of NIPT implementation using subgroup analysis and meta-regression accounted for some heterogeneity in the results looking at the uptake of IPD, namely the uptake of NIPT as a second line test and the risk threshold used for NIPT access. Other factors analysed didn't seem to account for this heterogeneity. Pooled estimates of key outcome measures of NIPT impact in populations around the world were provided, an important starting point for the continual monitoring of NIPT impact.

### Strengths

This systematic review followed focussed review questions, with inclusion and exclusion criteria selected *a priori* and published on the PROSPERO register. It utilised an exhaustive search strategy on multiple databases, using sources that will deliver both published materials and grey literature. A standardised risk of bias assessment tool was used, with appropriate subgroup, meta-regression and sensitivity analyses to explore sources of heterogeneity and bias in the included studies. The screening of manuscripts was undertaken by two independent reviewers, and discussion taken to a third reviewer for settling disagreements. The main outcomes of interest; uptake of NIPT, uptake of invasive prenatal diagnosis, termination of pregnancy, and live births of babies with Down's syndrome reflect relevant outcomes of high interest to policy makers, clinicians and families and we are able to provide important information on the overall trends in decision making within populations where NIPT has been introduced.

### Limitations

Not every study included in part A provided details of NIPT implementation or uptake values for their population, limiting the available evidence for comparison between screening programmes. Uptake data was also limited by the level of detail provided, for example with some articles not providing the denominator data for NIPT uptake.

This review is restricted to published material and grey literature referenced in included studies, and as government guidelines or recommendations may not be always made publicly available or published this will have limited the number of countries included in our results.

National policies on NIPT implementation that are not mentioned in peer reviewed literature will have been omitted.

The studies included in this review are mostly observational cohort studies. These are difficult to assess for quality, as many elements of the data collection are not able to be controlled. The Downs and Black scoring system has been used. However, as none of these studies are randomised cohort trials there will inherently be bias in these studies.

Many studies do not report the data necessary to analyse the outcomes of interest (missing data, lost to follow up, not enough data to estimate raw frequencies). Where data is reported, often the sample sizes were small and confidence intervals very wide. Pre-NIPT implementation data was not separately searched for, which means we have limited data to compare to available post-NIPT implementation data. Data on high chance women after biochemical screening was extracted so that it would be comparable between pre and post NIPT periods, where biochemical screening was used in both, but the raw data for women undergoing each type of screening was not often reported or available.

The meta-analyses showed high heterogeneity overall. Random effect models were used to adjust for statistical heterogeneity, and sensitivity and subgroup analyses to explore its possible sources.

## Comparison with existing literature

This review sought to provide an updated understanding of where and how NIPT has been implemented as part of a national screening programme. The results demonstrate that NIPT is available globally, although mostly among European and high-income countries. Changes and updates to the provision of NIPT were identified and may be due to the initial cost of the test gradually decreasing with the more common use and availability of this technology, which has meant many countries have been able to introduce this test as a first line screen [44].

The infrastructure and provision of healthcare resources, including cost and access, differ greatly between countries and will influence the acceptability of a new screening test or process such as NIPT. A comparative study of NIPT use in Quebec (Canada) and Lebanon highlighted the barriers to access and ethical considerations presented in each population. This included the cost of screening tests, coverage by insurance, lobbying by disability rights activists in Quebec, and the attitudes towards termination of pregnancy [45]. Similarly, the contrast between NIPT implementation in Germany and Israel has been discussed by Raz et al., 2021 [46], and the contrasting use of NIPT and invasive testing between Denmark and Israel [41]. These aspects were not able to be accounted for in analyses and could inevitably be influencing the variation seen in the results.

When analysing the uptake of screening, and the prevalence of live births and terminations of pregnancy within our included studies, it is important to consider the variation in legal and wider cultural factors between the countries/regions represented that may influence findings. For example, in S2 Table, we present the current termination of pregnancy laws for the areas represented in part A of our review. Substantial variation between areas under which termination is permitted, and related gestational limits, is evident. Although we are not able to formally investigate any specific association between termination of pregnancy laws and the quantitative outcomes measured in part B, it is likely that variation in legal and wider cultural factors explains, at least in part, the variation in outcomes and heterogeneity present in our meta-analyses.

Overall, vast heterogeneity in the healthcare systems and cultural differences between populations and regions will strongly influence the use and uptake of NIPT. What is clear is that each country has implemented NIPT in a different way, depending on several factors that might include the pre-existing infrastructure for a screening programme; how the healthcare

system runs; expected cost of healthcare; the cultural and societal expectations on pregnancy screening and acceptability of disability in these populations.

## Implications for clinical practice / policy

Numerous professional bodies have recommended the implementation of NIPT into screening programmes [47, 48] citing benefits such as cost effectiveness, improved accuracy for detecting trisomies, and a potential reduction in the number of invasive procedures performed. When considering the methods of implementation and eligibility criteria for access to NIPT there seems to be no consensus in how best to achieve this, instead pre-existing healthcare and societal aspects play significant roles.

Since first marketed in 2011, NIPT has expanded rapidly from the commercial to public sectors. As argued by Dougan et al. [26], the highly commercialised and marketed nature of the introduction of NIPT resulted in the quick, and unstandardised implementation of this test into the public sector. It is therefore important to understand the extent of availability and the impact this has had on the choices made by women and on the outcomes of babies with DS, warranting ongoing examination at a population level.

There is a lack of comprehensive data on the uptake of NIPT in populations, presented for a few countries included in this review. Moreover, there will be an inevitable lag between the implementation of a new technology and the publishing of any evaluation data, as well as many countries not having the resources to collect the required data to present these statistics. Consequently, data from those areas where NIPT has been more recently implemented will be unavailable, and subsequently the reported list of countries or states implementing NIPT will not be exhaustive, and the outcome data included in the meta-analyses limited.

The data analysed in this systematic review, and lack of completeness, emphasizes the requirement for further population-based data to be published, which should allow for comparison between a more diverse set of countries and possibly start to demonstrate associations between population characteristics and uptake of NIPT.

Being a particularly sensitive topic, many countries may be reluctant to publicly endorse pregnancy screening for DS or publish their guidelines; suggested as one of the reasons behind countries such as Germany not introducing a nation-wide policy on NIPT [46].

Ultimately, policy decisions impact on the choices made during pregnancy, and careful consideration should be given to the way these changes are implemented. Transparent and publicly available data is essential for a global approach to monitoring such impactful changes to screening programmes for DS.

In conclusion, NIPT has been implemented as an antenatal screening test in national DS screening programmes in many countries, and in a variety of ways, depending on the pre-existing healthcare structure and resources of that country. The uptake of NIPT is seen to differ greatly between populations, with no clear association with how NIPT has been implemented. There is evidence that the number of pregnancies undergoing invasive prenatal diagnosis has reduced after second line NIPT screening was implemented. The impact of NIPT on terminations of pregnancy and live births of babies with DS cannot be examined in this review as comparable data for pre-NIPT period is not available. Further studies using comparative pre and post NIPT data in the same populations are required to understand its impact on these outcomes.

## Supporting information

**S1 Fig. Stages of the routine screening pathway (biochemical screening) for Down's syndrome in the UK before the introduction of NIPT.** Also common to other countries with a routine screening programme. Timing of the screening tests, blood markers measured and

threshold for 'higher chance' pregnancy may vary between health systems. Invasive prenatal diagnosis = amniocentesis or chorionic villus sampling. * The higher chance threshold is calculated as 1 in X pregnancies, with thresholds being set for progression to invasive prenatal diagnosis. NT = nuchal translucency.
(TIF)

**S1 Table. Characteristics of included studies and search strategy.** Table 1 of all studies included in this systematic review, along with Down's and Black quality assessment score. 2. Search strategy, keywords and mesh terms used in database search.
(DOCX)

**S2 Table. Termination of pregnancy laws in included regions.**
(DOCX)

**S1 File. Copy of published PROSPERO study protocol.** Systematic review protocol document, registered on the PROSPERO database.
(PDF)

**S2 File. PRISMA checklist.** Completed PRISMA checklist for systematic review reporting, describing where each point is executed in the systematic review.
(DOCX)

**S3 File. Data analysis (part B).** Details of subgroup/meta-regression data analysis in part B of the systematic review.
(DOCX)

## Acknowledgments

This work was supported by the Medical Research Scotland PhD Studentship award [PHD-50200-2020].

## Author Contributions

**Conceptualization:** Elinor Sebire, Sohinee Bhattacharya, Rachael Wood.

**Data curation:** Elinor Sebire, Chithramali Hasanthika Rodrigo, Rute Vieira.

**Formal analysis:** Elinor Sebire, Rute Vieira.

**Investigation:** Elinor Sebire, Rute Vieira.

**Methodology:** Elinor Sebire, Chithramali Hasanthika Rodrigo, Sohinee Bhattacharya, Rute Vieira.

**Project administration:** Elinor Sebire.

**Supervision:** Sohinee Bhattacharya, Mairead Black, Rachael Wood, Rute Vieira.

**Visualization:** Elinor Sebire, Sohinee Bhattacharya, Mairead Black.

**Writing – original draft:** Elinor Sebire.

**Writing – review & editing:** Elinor Sebire, Chithramali Hasanthika Rodrigo, Sohinee Bhattacharya, Mairead Black, Rachael Wood, Rute Vieira.

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
