## [Decision Letter · Decision Letter 0]

21 Nov 2023

PONE-D-23-33164The implementation and impact of non-invasive prenatal testing (NIPT) for Down’s syndrome into antenatal screening programmes: a systematic review and meta-analysisPLOS ONE

Dear Dr. Sebire,

Thank you for submitting your manuscript to PLOS ONE. After careful consideration, we feel that it has merit but does not fully meet PLOS ONE’s publication criteria as it currently stands. Therefore, we invite you to submit a revised version of the manuscript that addresses the points raised during the review process.

We look forward to receiving your revised manuscript.

Kind regards,

Giuseppe Novelli

Academic Editor

PLOS ONE

Journal Requirements:

2. Please ensure that your PRISMA flow diagram is included in your main manuscript file as Figure 1; please see the PLOS ONE submission guidelines for systematic reviews and meta-analyses at https://journals.plos.org/plosone/s/submission-guidelines#loc-systematic-reviews-and-meta-analyses.

6. We note that Figure 3 in your submission contain map images which may be copyrighted. All PLOS content is published under the Creative Commons Attribution License (CC BY 4.0), which means that the manuscript, images, and Supporting Information files will be freely available online, and any third party is permitted to access, download, copy, distribute, and use these materials in any way, even commercially, with proper attribution. For these reasons, we cannot publish previously copyrighted maps or satellite images created using proprietary data, such as Google software (Google Maps, Street View, and Earth). For more information, see our copyright guidelines: http://journals.plos.org/plosone/s/licenses-and-copyright.

We require you to either present written permission from the copyright holder to publish these figures specifically under the CC BY 4.0 license, or (2) remove the figures from your submission:

a. You may seek permission from the original copyright holder of Figure 3 to publish the content specifically under the CC BY 4.0 license.  

Reviewers' comments:

Reviewer's Responses to Questions

**Comments to the Author**

1. Is the manuscript technically sound, and do the data support the conclusions?

Reviewer #1: Partly

Reviewer #2: Partly

2. Has the statistical analysis been performed appropriately and rigorously? 

Reviewer #1: Yes

Reviewer #2: N/A

3. Have the authors made all data underlying the findings in their manuscript fully available?

Reviewer #1: Yes

Reviewer #2: No

4. Is the manuscript presented in an intelligible fashion and written in standard English?

Reviewer #1: Yes

Reviewer #2: Yes

5. Review Comments to the Author

Reviewer #1: In the field of fetal DNA floating in maternal peripheral blood, the acronym NIPT has to be indicated at least once as cffDNA-NIPT in papers where various non-invasive testing methods are analyzed. The described background of this paper is the basis of all global government programs for the reduction of invasive genetic diagnosis for Down syndrome. In such global scenario the data evaluation criteria must include also the ethnic, legal and ethical differences that strongly influence the application of the tests and the decision-making aspect of the pregnant woman. The quantitative amount of the case-sample used for the analysis must be presented in the text with the numerical values of the cases alongside the percentages, referring only the details to the tables. The decision of the pregnant women in terms of therapeutic abortion or DS livebirts is not relevant for the aim of the study, so the number of abortions and affected births cannot be the subject of meta-analysis but only of analytical and documented description by patient's age, gestation period, legal regulations of the country. The data of 27 selected papers, a small number, is more eligible for a traditional review than a meta-analysis, to be able to appreciate and not lose specific characteristics of each study. The great amount of limitations of the study described by the authors in their conclusions demonstrates that the meta-analysis does not allow to produce added value to the study exceeding some intuitive practical conclusions.

Reviewer #2: In this study the authors do a systematic review to understand the extent of NIPT introduction for Down’s syndrome into national screening programmes worldwide, its uptake among eligibile populations and the impact it may have on specific pregnancy outcomes.

As reported by the authors themselves, this review has many limitantions because many studies used for the meta-analyses do not report all the data necessary to analyse the outcomes of interest.

Furthermore, the paper has some points to be reviewed.

1. Some points regarding NIPT need to be corrected:

- DNA sequencing methods are not the only technologies used in non-invasive prenatal testing (lane 12-13), there are other technologies such as microarrays, nanofiltration plates with fluorescence scanning, Droplet

Digital PCR (ddPCR)

- re-evaluate literature reference 6 in lane 58;

- false-positive cases may be indicative of placental mosaicism, missing twins and "presence of maternal chromosomal abnormalities", the latter condition may be suggestive of maternal cancer, but not only (lane 71).

2. In Figure 2 and in the text, the authors do not explain all the reasons why 125 studies out of 167 were excluded from the review (30 are excluded due to 'unsuitable study population', 42 due to 'experimental study design' and the remaining 53?).

3. In Table 3, the authors refer to the Gadsboll et al. 2020 paper, but do not detail the situation in the individual countries reported as they should. For example, in Italy there are official guidelines supporting the use of NIPT in high-risk women already before 2019, but only some regions, such as Tuscany or Apulia, currently reimburse the test.

6. PLOS authors have the option to publish the peer review history of their article (what does this mean?). If published, this will include your full peer review and any attached files.

Reviewer #1: No

Reviewer #2: No

---

## [Author Response · Author response to Decision Letter 0]

19 Dec 2023

Dear Editor,

Thank you for the opportunity to revise and resubmit our manuscript ‘The implementation and impact of non-invasive prenatal testing (NIPT) for Down’s syndrome into antenatal screening programmes: a systematic review and meta-analysis’ for publication in PLOS ONE. We appreciate the time dedicated by you and the reviewers to provide feedback on the manuscript, offering insightful comments and valuable improvements to make. We have revised the manuscript to reflect the reviewers’ comments, which are sign-posted in the track changes version of the manuscript attached. We have provided a point-by-point response to each of the reviewers' comments which can be found in the 'Response to Reviewers' document attached with the revised submission.

---

## [Editor Report · Decision Letter 1]

29 Jan 2024

The implementation and impact of non-invasive prenatal testing (NIPT) for Down’s syndrome into antenatal screening programmes: a systematic review and meta-analysis

PONE-D-23-33164R1

Dear Dr. Sebire,

We’re pleased to inform you that your manuscript has been judged scientifically suitable for publication and will be formally accepted for publication once it meets all outstanding technical requirements.

Kind regards,

Giuseppe Novelli

Academic Editor

PLOS ONE
---

## [Editor Report · Acceptance letter]

2 May 2024

PONE-D-23-33164R1 

PLOS ONE

Dear Dr. Sebire, 

I'm pleased to inform you that your manuscript has been deemed suitable for publication in PLOS ONE. Congratulations! Your manuscript is now being handed over to our production team.

Kind regards, 

on behalf of

Prof. Giuseppe Novelli 

Academic Editor

PLOS ONE